# Maize Apoplastic Fluid Bacteria Alter Feeding Characteristics of Herbivore (*Spodoptera frugiperda*) in Maize

**DOI:** 10.3390/microorganisms10091850

**Published:** 2022-09-16

**Authors:** Sellappan Ranjith, Thangavel Kalaiselvi, Muruganagounder Muthusami, Uthandi Sivakumar

**Affiliations:** 1Department of Agricultural Microbiology, Tamil Nadu Agricultural University, Coimbatore 641003, India; 2Department of Agricultural Entomology, Tamil Nadu Agricultural University, Coimbatore 641003, India

**Keywords:** maize, apoplastic fluid bacteria, fall armyworm, bioprotectant, bioassay

## Abstract

Maize is an important cereal crop which is severely affected by *Spodoptera frugiperda*. The study aims to identify endophytic bacteria of maize root and leaf apoplastic fluid with bioprotective traits against *S. frugiperda* and plant growth promoting properties. Among 15 bacterial endophytic isolates, two strains—namely, RAF5 and LAF5—were selected and identified as *Alcaligenes* sp. MZ895490 and *Bacillus amyloliquefaciens* MZ895491, respectively. The bioprotective potential of *B. amyloliquefaciens* was evaluated through bioassays. In a no-choice bioassay, second instar larvae of *S. frugiperda* fed on *B. amyloliquefaciens* treated leaves (B+) recorded comparatively lesser growth (1.10 ± 0.19 mg mg^−1^ day^−1^) and consumptive (7.16 ± 3.48 mg mg^−1^ day^−1^) rates. In larval dip and choice bioassay, the same trend was observed. In detached leaf experiment, leaf feeding deterrence of *S. frugiperda* was found to be greater due to inoculation with *B*. *amyloliquefaciens* than *Alcaligenes* sp. The phenolics content of *B. amyloliquefaciens* inoculated plant was also found to be greater (3.06 ± 0.09 mg gallic acid g^−1^). However, plant biomass production was more in *Alcaligenes* sp inoculated treatment. The study thus demonstrates the potential utility of *Alcaligenes* sp. and *B. amyloliquefaciens* for improving growth and biotic (*S. frugiperda*) stress tolerance in maize.

## 1. Introduction

Maize (*Zea mays*), the third most important cereal crop in India after rice and wheat, has been planted on about 80 lakh ha [1]. *Spodoptera frugiperda* is one of the severe pests of maize and causes yield losses of 8.3 to 20.6 million tons per annum. This is a devastating insect native to tropical and subtropical regions of America. This insect feeds on more than 80 plant species—including maize, rice, sorghum, millet, sugarcane, vegetable crops, and cotton, among others. In the recent past, it has been reported that maize is being affected heavily by *S. frugiperda* [2]. In the current scenario of climate change and global warming, it is important to identify bioprotective agents for the control of this herbivore.

Results of plant microbiome studies revealed that plants are closely associated with numerous beneficial microorganisms. These microbes are of immense attention because of their role in improved plant growth and health [3]. Plant endophytes, microbes inhabiting plant tissues, have been reported to enhance plant growth by improving nutrient availability and providing tolerance against biotic and abiotic stresses [4]. The mechanisms of improvement in plant health include triggering immune system via induced systemic resistance and through production of hydrogen cyanide [5], siderophores [6], ammonia [7], and hydrolytic enzymes such as proteases, chitinases, lipases, and pectinases [8]. The endophytes mediated priming and/or eliciting defence against various phytopathogenic fungi [9], herbivores [10], nematodes [11], and viruses [12] were reported in earlier studies. A plant’s apoplast is the space outside the plasma membrane which contains free diffusing metabolites and proteins. These solutes play an important role in plant physiology by controlling biotic and abiotic stresses. The apoplast is considered as one of the main reservoirs of bacterial endophytes and also it is an important site of interaction between beneficial endophytic microbes and external elicitors such as pathogens [13]. The apoplastic fluid endophytes improve plant growth through enhanced nutrient availability (nitrogen, phosphate, potassium, iron, and zinc) and phytohormone production (indole acetic acid and gibberellic acid) [10,14]. Earlier studies reported that apoplastic microbes could enhance plant growth, mitigate drought [15], and trigger immunity against phytopathogens [16].

Despite reports available for endophytic bacteria induced resistance against diseases and pests [10], the effect of apoplastic endophytic bacteria mediated defence priming against herbivores are poorly understood. In this context, the study was carried out with the objectives to isolate and screen potential plant growth promoting bacterial endophytes from apoplastic fluid of maize leaf and root and to evaluate the potential of selected endophytic bacterial inoculation on the growth of maize and *S. frugiperda* during herbivore (*S*. *frugiperda*) attack.

## 2. Materials and Methods

### 2.1. Isolation and Characterization of Apoplastic Fluid Endophytes from Maize (COH6)

The apoplastic fluid was recovered from excised roots and leaf samples of 45 days old maize COH6 grown in millet farm, Tamil Nadu Agricultural University (TNAU), Coimbatore, India according to the procedure of Maksimovica et al. [17]. In brief, samples were washed with tap water and cut into 5 cm pieces and soaked in 0.05% (*v*/*v*) triton × 100 for 4 min. After that, the samples were immersed in 5% sodium hypochlorite for 10 min, followed by 2.5% (*w*/*v*) sodium thiosulphate for 10 min; then washed several times with sterile distilled water. After that, the leaf segments were dipped in 70% ethanol for 10 min followed by ten times wash in sterile distilled water. The leaves were blot dried on sterile filter paper and imprinted on tryptic soy agar medium to check for contamination. For apoplastic fluid extraction, the samples were cut into 2 cm and placed in the sterile disposable syringe containing infiltration solution (100 mM KCl) followed by imposing pressure on samples until it turned a dark colour. The samples were placed inside 1 mL micropipette tip. The tip was placed in the falcon tube and centrifuged at 6000 g for 10 min. The filtered white apoplastic fluid was stored at 4 °C until further use. The collected apoplastic fluid (10 µL) was spread on nutrient agar (NA), lysogeny agar (LA), tryptic soy agar (TSA), and Reasoner’s 2A agar (R2A) with different concentrations (100%, 50%, and 25%) for 48 h at 37 °C. Morphologically distinct colonies from each media were selected.

### 2.2. Plant Growth Promoting Potential of Apoplastic Fluid Endophytes

#### 2.2.1. Estimation of Mineral Solubilisation Efficiency

For estimating nutrient solubilising index, the bacterial cultures were spotted on Pikovskaya, Aleksandrov, and Bunt and Rovira media for phosphate, potassium, and zinc respectively. After incubation, the halo zone formed around the colonies was measured. The mineral solubilising index was calculated according to Chakdar et al. [18].
(1)$$Mineral\;solubilisation\;index\;\left(SI\right)=\frac{Colony\;diameter+Halo\;zone\;diameter}{Colony\;diameter}$$

#### 2.2.2. Indole Acetic Acid (IAA)

IAA production by the apoplastic fluid endophytes was determined using Salkowski reagent following the procedure of Patel and Saraf [19]. Isolates were inoculated into LB broth containing 0.2% L-tryptophan, pH 7.0, and incubated at 28 °C for 7 days. Post incubation cultures were centrifuged at 12,000× *g* for 15 min. One mL of the supernatant was mixed with 2 mL of Salkowski reagent, and the intensity of the pink colour developed was read at 530 nm.

#### 2.2.3. Gibberllic Acid (GA)

Selected endophytes were first grown in 10 mL nutrient broth for seven days and then centrifuged at 12,000× *g* for 10 min. To the supernatant, 2 mL zinc acetate was added and incubated for 2 min. Then 2 mL potassium ferrocyanide was added and the mixture was centrifuged at 10,000× *g* for 10 min. To the 5 mL supernatant, 5 mL of 30% hydrochloric acid was added and incubated at 30 °C for 75 min [20]. After incubation, the absorbance was taken at 254 nm and various concentrations of GA were used for preparation of standard.

### 2.3. Analysis of Bioprotective Properties of Maize Apoplast Endophytes

#### 2.3.1. Siderophore Production

Chrome azurol succinic (CAS) acid medium was used for qualitative assay of siderophore production. The endophytes were streaked on CAS medium and change of the colour of the medium from green to yellow was considered positive for siderophore production. For quantitative assessment, 48 h old cultures grown in succinic acid broth were centrifuged at 10,000× *g* for 10 min. One mL supernatant was mixed with 1 mL of CAS and absorbance was taken at 630 nm in spectrophotometer (Spectramax^®^ i3X, San Jose, CA, USA) [12].

#### 2.3.2. Ammonia Production

The endophytes were inoculated into 10 mL peptone water broth and incubated at 30 °C for 72 h. After incubation, the broth was centrifuged at 10,000× *g* for 10 min. Subsequently, 1 mL supernatant was mixed with 0.5 mL Nessler’s reagent and the colour developed was measured at 450 nm in spectrophotometer. The quantity of ammonia produced was calculated using the known concentration of ammonium sulphate as standard [21].

#### 2.3.3. Hydrogen Cyanide (HCN) Production

HCN production was determined with the methodology of Devi and Thakur [22], using sodium picrate as an indicator. First, the selected endophytes were grown in a conical flask containing nutrient broth supplemented with glycine (4.4 g L^−1^). Then, Whatman No.1 filter paper (1 × 5 cm) strip dipped in sodium picrate solution was hanged in the conical flask and incubated at room temperature for 72 h. Upon incubation, the colour of the sodium picrate in the filter paper changed from yellow to reddish brown which is proportional to the concentration of HCN produced by the culture. The compound in the filter paper was eluted using 10 mL distilled water and absorbance was recorded at 625 nm.

#### 2.3.4. Lipase Activity

Isolated endophytes were streaked on tributyrin agar medium and incubated at 35 °C for 48 h. The clear zone around the colony indicated positive for lipase activity [23].

#### 2.3.5. Protease Activity

The bacterial endophytes were streaked on skimmed milk agar medium and incubated at 30 °C for 48 h. The endophytes positive for protease production were detected from formation of clear zone around the colony. To quantify the protease production, 24 h grown endophytes were centrifuged at 10,000× *g* for 6 min and the supernatant (crude enzyme) was collected. To 200 µL of the supernatant, 500 μL of casein (1% *w*/*v* in 50 mM phosphate buffer, pH-7) was added and incubated in water bath at 40 °C for 20 min. Then, 1mL of 10% (*w*/*v*) trichloroacetic acid was added and incubated at 35 °C for 15 min. The mixture was centrifuged at 10,000× *g* for 5 min and the supernatant was collected. To the supernatant, 2.5 mL of 0.4 M Na_2_CO_3_ and 1 mL of Folin-Ciocalteu’s phenol reagent were added and incubated at room temperature for 30 min in the dark. The optical density of the solution was read at 660 nm [24].

#### 2.3.6. Pectinase

The endophytes were streaked on pectinase agar medium and incubated for 48 h at 30 °C. Then, the culture plates were flooded with 50 mM potassium iodide solution. Appearance of clear zone around the colony indicated positive result for pectinase production. For quantitative analysis, the positive cultures grown in nutrient broth for 48 h were centrifuged at 10,000× *g* for 5 min and the supernatant served as a source of crude enzyme. To 100 µL of supernatant, 900 µL of substrate (citrus pectin (0.5% *w*/*v*) in 0.1 M phosphate buffer, pH-7.5) was added and incubated at 50 °C for 10 min in water bath. After incubation, 2 mL of dinitrosalicylic acid reagent (DNS) was added and placed in boiling water bath for 10 min. After cooling, optical density (OD) of the solution was measured at 540 nm [25].

#### 2.3.7. Chitinase

Isolated endophytes were streaked on colloidal chitin agar medium and incubated for 5 days at 35 °C and clear zone formation around the colony indicated positive result for chitinase activity [26]. The positive cultures grown in nutrient broth at room temperature for 48 h were centrifuged at 10,000× *g* for 15 min at 4 °C and the supernatant (crude enzyme) was collected. 150 µL crude extract was added to 150 µL of 0.1 M phosphate buffer (pH 7) and 300 µL of 0.1% colloidal chitin. Then incubated at 55 °C for 10 min; the mixture was centrifuged at 10,000× *g* for 5 min and the supernatant was collected. Then, 200 µL supernatant was mixed with 0.5 mL distilled water and 1 mL of Schales reagent; the mixture was boiled for 10 min and the absorbance was recorded at 420 nm.

### 2.4. Molecular Identification of Potent Apoplastic Fluid Bacterial Endophytes

Among 15 endophytes, two endophytes were selected based on the growth promotions characteristics and bioprotective properties. The selected endophytes grown overnight in nutrient broth were used for extraction of genomic DNA following CTAB (cetyltrimethylammonium bromide) method [27]. 16S rRNA sequences was amplified using, 27F (5′AGAGTTTGATCCTGGCTCAG3′) and 1492 R (5′GGTACCTT GTTACGACTT3′) primers in PCR (polymerase chain reaction) thermocycler (Biorad T100). The PCR product was sequenced (J.K. Scientific, Coimbatore, India) and submitted in National Centre for Biotechnology Information (NCBI).

### 2.5. Effect of Bacillus amyloliquefaciens on the Leaf Feeding Capacity of S. frugiperda Larvae

#### 2.5.1. Insect, Crop, and Endophyte Used for the Study

*Spodoptera frugiperda* egg mass procured from National Bureau of Agricultural Insect Resources (NBAIM), Bangalore (Accession number: NOC-03) was reared in plastic container with maize leaves as feed till they reached second instar [28]. Maize (COH6) seeds were collected from Department of Millets, Tamil Nadu Agricultural University, Coimbatore. The apoplastic endophyte *B. amyloliquefaciens* was grown in nutrient broth at 35 °C for 24 h and then centrifuged at 10,000× *g* for 10 min and the cell pellet was collected. The cell concentration was adjusted to 10^8^ cfu mL^−1^ with sterile distilled water.

#### 2.5.2. No-Choice and Choice Bioassay

The bioprotective potential of *B. amyloliquefaciens* against *S. frugiperda* was determined through bioassays. For the bioassays, 5 cm maize (COH6) leaves (35 days old plants from Millet field, TNAU) were soaked in sterile water containing 0.001% Tween 80 for 10 min and washed five times with sterile distilled water. Then leaves were surface sterilized with 1.6% sodium hypochlorite for 3–4 min and washed five times with sterile distilled water. In no-choice bioassay, the surface sterilized pre-weighed leaves were dipped in 20 mL of *B. amyloliquefaciens* (10^8^ cfu mL^−1^) (B+) for 15 min, simultaneously control leaves (B−) were soaked in sterile distilled water. Excess inoculum was drained off and each treated leaf was placed on a sterile petri plate containing wet filter paper. Then, a pre-weighed second instar larva was released into the petri plate and allowed to feed for 24 h. In choice bioassay, two leaves—one *B. amyloliquefaciens* inoculated (BA+) and another un-inoculated (BA−)—were placed in a petri plate (140 mm). One pre-weighed second instar larva was released, allowed to feed both the leaves and observed for the preference [29]. Each bioassay was carried out with 25 replicates. The feeding capacity of larva was evaluated statistically based on the methodology of Waldbauer [30].
(2)$$Relative\;growth\;rate\;\left(RGR\right)=\frac{Weight\;gain}{Initial\;weight\;of\;caterpillar}$$
(3)$$Relative\;consumptive\;rate\;\left(RCR\right)=\frac{Food\;consumption}{Initial\;weight\;of\;caterpillar}$$
(4)$$Efficiency\;of\;conversion\;of\;ingested\;food\;\left(ECI\right)=\frac{B}{I}\times 100$$
(5)$$Efficiency\;of\;conversion\;of\;ingested\;food\;\left(ECI\right)=\frac{B}{I}\times 100$$
(6)$$Efficiency\;of\;conversion\;of\;digested\;food\;\left(ECD\right)=\left(\frac{B}{I}-F\right)\times 100$$
(7)Metabolic cost (MC)=100−ECD
(8)$$Feeding\;deterrent\;index\;\left(FDI\right)=\frac{C-T}{C+T}\times 100$$
where *B*—larval weight gain, *I*—food ingested, *C*—weight of control leaf, *T*—weight of treated leaf.

#### 2.5.3. Larval Dip Bioassay

In this assay, second instar larvae (*n* = 25) were soaked in 20 mL *B. amyloliquefaciens* culture (10^8^ cfu mL^−1^) for 5 s. For control, the same numbers of larvae were dipped in sterile distilled water [31]. One surface sterilized maize (COH6) leaf segment (from 35 days old plant) was placed in petri plate containing (90 mm) sterile wet filter paper. Then, the second instar larva was introduced and allowed to feed for 24 h. The feeding capacity of larva was calculated [30].

### 2.6. Effect of B. amyloliquefaciens and Alcaligenes sp. Treated Maize (COH6) on S. frugiperda Growth

#### Detached Leaf Bioassay

A pot culture experiment with completely randomized design was conducted in the Department of Agricultural Microbiology, TNAU, Coimbatore. Treatments comprised of T1—Control without *Alcaligenes* sp., *B. amyloliquefaciens* and with *S. frugiperda* (C*SF); T2—*Alcaligenes* sp. + *S. frugiperda* (AS*SF); T3—*B. amyloliquefaciens* + *S. frugiperda* (BA*SF); T4—*Alcaligenes* sp. + *B. amyloliquefaciens* + *S. frugiperda* (AS*BA*SF). Surface sterilized (10% sodium hypochlorite for 10 min) maize (COH6) seeds were soaked in 15 mL (10^8^ cfu mL^−1^) of *Alcaligenes* sp. which was isolated from root apoplastic fluid and incubated for overnight. Then, the maize seeds were sown in mud pots (22 × 20 × 20 cm) containing sterile soil and sand (2:1). Five days after seed germination, 3 mL of 24 h culture of *B. amyloliquefaciens* (10^8^ cfu mL^−1^) was sprayed on foliar region of each plant using low volume sprayer. Hoagland nutrient (100 mL pot^−1^) was poured once in five days after germination of seeds. The pots were irrigated once in two days with 100 mL tap water pot ^−1^. After 35 days of germination, leaf samples (*n* = 10) were collected from each treatment and cut into 5 cm segments. For the bioassay, single leaf segment was placed in a petri plates containing wet filter paper and one pre-weighed second instar *S. frugiperda* (SF) larva was introduced for 24 h feeding [32]. The feeding capacity of larva was calculated [30].

### 2.7. Changes in Phenolics Content and Dry Matter Production of Endophytes (B. amyloliquefaciens and Alcaligenes sp.) Inoculated Maize during S. frugiperda Infestation

#### 2.7.1. Experimental Design

A pot experiment with four treatments, T1—Control without Alcaligenes sp., *B. amyloliquefaciens* and with *S. frugiperda* (C*SF), T2—Alcaligenes sp. + *S. frugiperda* (AS*SF), T3—*B. amyloliquefaciens* + *S. frugiperda* (BA*SF), T4—Alcaligenes sp. + *B. amyloliquefaciens* + *S. frugiperda* (AS*BA*SF) was conducted with five replication as mentioned in the Section 2.6. After 35 days of germination, two second instar larvae were introduced in a plant for 24 h feeding.

#### 2.7.2. Phenolics Content

After herbivory treatment, leaf samples were drawn, and 500 mg of the sample were ground with 1 mL of 80% methanol and centrifuged at 10,000× *g* at 4 °C for 10 min and the supernatant was collected. The phenolics content in the supernatant was estimated as per the protocol of Selvaraj et al. [32]. Accordingly, 0.2 mL supernatant was mixed with 1 mL of 1 N Folin-Ciocalteu reagent and 1 mL distilled water and incubated for 3 min at 30 °C. After incubation, 1 mL of 20% sodium carbonate was added, and incubated in water bath (100 °C) for a minute and then cooled. After cooling, the absorbance was measured at 725 nm.

#### 2.7.3. Plant Biomass Production

After 24 h of herbivore treatment, the dry weight of shoot and root were recorded and expressed as g plant^−1^.

### 2.8. Statistical Analysis

Statistical analysis was carried out with the software, XLSTAT (version 2019.2.1) and SPSS (version 16.0). The values are represented as mean ± standard error of experimental data with minimum of three replications. The Duncan’s multiple range test (DMRT) was performed at *p* ≤ 0.05 for all the endophytic characterization, detached leaf bioassay experiments. Principal component analysis (PCA) was performed to identify the best bioprotective apoplastic endophytic bacteria. Independent sample *t*-test was done for the data concerned with no choice and larval dip bioassay. The Box and Wishker plots were carried out for the phenolics test.

## 3. Results

### 3.1. Isolation of Endophytes from Leaf and Root Apoplastic Fluid of Maize (COH6)

Totally 15 bacterial endophytes were isolated from maize leaf and root apoplastic fluid. Of 15, nine bacterial endophytes were isolated from leaf apoplast (LAF—Leaf apoplastic fluid) and six were obtained from root apoplast (RAF—Root apoplastic fluid) using different media with various concentrations (Appendix A). The morphological characteristics of these isolates were studied and represented in Appendix A. These 15 endophytes were characterized for their plant growth promoting activities and bioprotective properties.

### 3.2. Growth Promoting Characteristics of Isolated Maize Apoplastic Fluid Bacterial Endophytes

The potential mineral solubilizing endophytes were selected based on their mineral solubilisation index (SI). All the three minerals were solubilized by six endophytes (LAF2, LAF7, RAF3, RAF4, RAF5, and RAF6). Among them, RAF5 showed significantly (*p* = 0.001, df = 12, Table 1) higher degree of mineral solubilisation. LAF8 showed phosphate and potassium solubilisation, whereas RAF2 solubilized phosphate and zinc only. GA and IAA production were noticed in all the 15 endophytes. Of the 15 isolates, RAF5 produced a significantly (*p* = 0.001, df = 30, Table 1) higher amount of IAA (58.04 ± 0.07 µg mL^−1^) and GA (30.61 ± 0.02 µg mL^−1^) than other endophytes.

### 3.3. Screening Bacterial Endophytes for Their Bioprotective Potential

Among 15 endophytes, 10 isolates exhibited positive results for siderophore, hydrogen cyanide (HCN) and ammonia production. Of these 10 isolates, RAF2 (12.29 ± 0.04%) recorded maximum siderophore production; RAF5 recorded maximum HCN production and LAF5 (1.68 ± 0.03 µg mL^−1^) recorded maximum ammonia production (Table 2). In case of hydrolytic enzymes, the endophytes LAF5 and LAF7 were found to produce lipases, proteases, pectinases, and chitinases. While LAF6 showed positive for lipases, proteases, and pectinases tests (Table 2).

### 3.4. Molecular Identification of Potent Bacterial Endophytes

Based on the plant growth promotion activities and bioprotective properties, RAF5 (Table 1) and LAF5 (Table 2, Figure 1) were selected and sequenced. The nucleotide sequences were analyzed using BIOEDIT software resulted in contigs of 1445 bp (RAF5) and 710 bp (MAL5). The NCBI blast analysis of RAF5 (1445 bp) showed higher homology with *Alcaligenes* sp. with sequence similarity of 100% and e-value 0. LAF5 (710 bp) showed higher similarity with *B. amyloliquefaciens* with sequence similarity of 100% and e value 0. The nucleotide sequences of these endophytes were submitted in the NCBI, and the accession numbers obtained were MZ895490 for RAF5 and MZ895491 for LAF5 (Figure 2).

### 3.5. Evaluation of Bioprotective Potentiality of B. amyloliquefaciens against S. frugiperda in Maize

#### 3.5.1. No-Choice Bioassay

The larvae fed with *B. amyloliquefaciens* treated leaves showed significant reduction in their relative growth rate (RGR) (*p* = 0.001, t = 18.65, Table 3). The relative consumptive rate (RCR) of larvae also reduced due to feeding of *B. amyloliquefaciens* treated leaves (BA+) (*p* = 0.001, t = 216.27). However, the efficiency of conversion of ingested food was higher in larvae fed on BA+ (13.97 ± 6.35%, *p* = 0.042, t = 36.51) leaves than in B− (8.26 ± 1.73%). Similarly conversion efficiency of digested food was higher in BA+ (49.29 ± 2.01%, *p* = 0.001, t = 145.66) over BA− (21.88 ± 1.42%). Hence, the metabolic cost was lower in BA+ (50.71 ± 0.29%, *p* = 0.000, t = 25.33) and higher in BA− (78.12 ± 0.56%). However, the feeding deterrent index was higher (42.64 ± 2.64%, *p* = 0.032, t = 587.88) when fed on BA+ leaves.

#### 3.5.2. Choice Bioassay

The results of choice bioassay showed that the percentage preference by the larvae for feed was 35% for BA− and 65% for BA+. However, among 65%, 20% of larvae shifted their preference from BA+ to BA−. The mortality rate due to feeding of BA+ leaves was 15%. The larval weight was greater when fed on BA− (1.83 ± 0.23 mg g^−1^ day^−1^) compared to BA+ (0.90 ± 0.41 mg g^−1^ day^−1^).

#### 3.5.3. Larval Dip Bioassay

The relative growth rate (*p* = 0.02, t = 18.52) and the relative consumptive rates (*p* = 0.001, t = 22.04, Table 4) of larvae fed on BA+ leaves were significantly lower than those of larvae fed on BA− leaves (Table 4). However, the rates of conversion efficiency of ingested (*p* = 0.042, t = 48.26) and digested food (*p* = 0.001, t = 37.54) were greater for BA+ fed larvae. The metabolic cost was lower in BA− than BA+ (*p* = 0.001, t = 18.50). However, the feeding deterrent index of larvae was (17.70 ± 6.62%, *p* = 0.032, t = 86.45) significantly higher in BA+.

### 3.6. Effect of Alcaligenes sp. and B. amyloliquefaciens Inoculated Maize as Feed on the Growth of S. frugiperda

#### 3.6.1. Detached Leaf Bioassay

Leaf feeding bioassay indicated that the relative growth rate of *S. frugiperda* (SF) issignificantly higher when fed on AS*SF (1.63 ± 031 mg mg^−1^ day^−1^) (*p* = 0.04, df = 27, Table 5) and lower in BA*SF (0.60 ± 0.15 mg mg^−1^ day^−1^). The relative consumptive rate was significantly higher in C*SF (9.55 ± 5.63 mg mg^−1^ day^−1^) and recorded lower in BA*SF (2.09 ± 0.49 mg mg^−1^ day^−1^). However, conversion efficiency of ingested food was higher in BA*SF (30.00 ± 6.13%) followed by AS*SF (24.53 ± 1.26%). Comparatively, the efficiency of conversion of digested food was lower in BA*SF (3.89 ± 0.17%). It was followed by AS*SF (5.95 ± 0.22%). Although the metabolic cost was higher in BA*SF (96.11 ± 0.17%), it was not significantly (*p* = 0.073) different from other treatments. However, the feeding deterrent index was significantly higher in BA*SF (68.00 ± 2.69%).

#### 3.6.2. Phenolics Content of Maize Leaves

The phenolics content of BA*SF inoculated maize (3.06 ± 0.12 mg gallic acid g^−1^) was significantly higher than other treatments (*p* = 0.02, df = 12, Figure 3) except AS*BA*SF (*p* = 0.09, t = 0.89). However, there was no significant difference between C*SF and AS*SF (*p* = 0.735, t = 0.363).

#### 3.6.3. Biomass Content of Maize

Endophytes inoculated maize recorded higher biomass than uninoculated control against *S. frugiperda* (Figure 4). Among the endophytes, AS*SF treated plants registered significantly (*p* = 0.001, df = 32) higher shoot biomass content except BA*SF (*p* = 0.12, t = 1.84). Similarly, root biomass was also significantly higher in AS*SF (*p* = 0.001, df = 32) inoculated plants. Root biomass value was lower in AS*BA*SF compared to treatments like AS*SF and BA*SF; however, it was significantly higher than control (*p* = 0.003).

## 4. Discussion

The apoplast is the region exterior to the protoplasm wherein lot of interactions occurs and is the main reservoir for bacterial endophytes. It is the space in which microorganisms interact with each other and also with the host. Thus, the present study was aimed to identify potential bacterial endophytes from apoplastic fluid to promote maize growth and protect the host from the most dangerous herbivore, fall armyworm (*S. frugiperda*). Plant growth promoting bacterial (PGPB) endophytes improve crop growth by enhancing nutrient availability (nitrogen, phosphorus, potassium, sulphur, zinc, and iron), providing plant growth hormones (IAA, GA, cytokinins) [33], protecting plants from pests and diseases [7] and imparting tolerance against biotic and abiotic stressors [11,15]. In the current study, 15 endophytes were obtained from maize COH6 leaf and root apoplastic fluid. They were characterized for plant growth promoting and bioprotecting properties. Among 15 isolates, two endophytes were found to be superior to others. These two endophytes were identified phylogenetically as *Alcaligenes* sp. and *B. amyloliquefaciens.* Among these two isolates, *Alcaligenes* sp. (MAR5) showed significant potentiality for all the plant growth promoting properties (Table 1, Figure 4). Similarly, *Alcaligenes feacalis* inoculated rice recorded more germination percentage, biomass content, and root and shoot length than the control [34]. A plethora of research cited the importance of *Alcaligenes* sp. in plant growth promotion and stress alleviation [35].

To improve the resistance and/or tolerance of maize against *S. frugiperda*, the biochemical and physiological properties (siderophore, HCN, ammonia) of the maize apoplastic fluid endophytes were tested and these compounds were also reported to promote plant growth. Production of siderophore (as ligands to solubilize the iron) enhance the defence signalling of plants against various pathogens and pests. Excess intake of iron led to iron toxicosis and results in insect mortality [36]. The hydrogen cyanide (HCN) is a secondary metabolite produced by bacteria and has potential effect on external pathogen elicitors. The HCN disrupts the electron transport chain, which causes cell death. The HCN producing *Chromobacterium* sp. affected the growth of *Anopheles gambiae* mosquito larvae and increased mortality rate [37]. Ammonia is a volatile compound which is used for plant growth and disease suppression. Pal et al. [38] reported that ammonia producing *Enterobacter cloacae* control *Pythium ultimum* (damping off) in cotton and in the current study also most of the bacterial species produced siderophore, HCN, and ammonia.

Bioprotective properties also depend on the capability of endophytes to produce hydrolytic enzymes such as lipases, proteases, pectinases, and chitinases. Lipases hydrolyse waxes, lipoproteins, and fat in the insects; proteases affect the insect cuticles, midgut, and hemocoel; chitinases disrupt the cell wall cuticle and pectinases also have the role in pest control which affects insect gut [39]. Based on these characteristics LAF5 (*B. amyloliquefaciens*) was chosen as the better endophyte among other apoplastic isolates (Table 2, Figure 1).

Plethora of references reported the importance of *Bacillus* spp. in plant growth promotion and biocontrol properties which include *B. amyloliquefaciens* from canola which possessed multiple biocontrol traits such as siderophore, HCN, and ammonia production and the bacterium was shown to be antagonistic to pathogen [5]. Similarly, the endophytic *Bacillus* sp. was found to produce substantial amounts of proteases and lipases [39]. The chitinase produced by *Bacillus* spp. was found to induce defence genes and enzymes during biotic stress in tomato [40]. Similarly, the current investigation showed protease, lipase, and chitinase production by *B. amyloliquefaciens* (Table 2). Numerous references are available for supporting bioprotective property of *B. amyloliquefaciens* against phytopathogens [41]. *B. amyloliquefaciens* strain C6c capable of secreting hydrolytic enzymes was shown to be antagonistic to pathogens which colonize leaves, seed, and stem of English ivy [42]. Similarly, a lower degree of anthracnose infection was noticed in tobacco plants inoculated with *B. amyloliquefaciens* compare to control [43]. Although plenty of reports indicate the effect of *Bacillus* sp. against plant pathogens, reports are scant for *Bacillus* sp. mediated protection of plants from herbivorous insects attack. Some reports [44] showed high mortality rate of spotted stem borer (*Chilo partellus*) due to feeding of endophytic *Bacillus* sp. inoculated maize indicate the possibility of utilizing bacterial endophytes for protecting crops from herbivorous insect attack. Inoculation and colonization of broad bean with *B. amyloliquefaciens* significantly reduced the feeding capacity of aphids through defence priming [43]. Hence, in the current study, we investigated the importance of *Bacillus amyloliquefaciens* in protecting maize form *S. frugiperda* attack.

In no choice bioassay, *B. amyloliquefaciens* dipped leaves fed larvae showed lesser growth compared to control (Table 3). In larval dip method also growth and consumption rate of *B. amyloliquefaciens* treated larvae reduced significantly (Table. 4). Similarly, Kaushik et al. [45] reported that *S. frugiperda* showed lower growth rate when fed with fungal endophytes inoculated grass than un-inoculated control. Clement et al. [46] reported that *Rhopalosiphum padi* aphids choose endophyte free grass compare to endophyte infected grass in choice test. Crawford et al. [47] also observed that both, in field and pot trails, herbivores significantly preferred endophyte free grasses. On the contrary, in our study, most of the larvae (65%) preferred *B. amyloliquefaciens* inoculated leaves (BA+) initially. However, they have changed their preference from inoculated to un-inoculated leaves (BA−). Nearly 20% of larvae changed their option from inoculated (BA+) to un-inoculated leaves (BA−) and 15% larvae fed on treated leaves died.

The detached leaf bioassay indicated that the highest relative growth and consumptive rates of larvae were found in control plants over treated. However, the conversion efficiency and digestibility of feeding material were recorded in higher percentages in endophytes applied treatments than in the control. These observations are similar to the result for *Spodoptera litura* fed on black gram inoculated with AMF and *Rhizobium* by Selvaraj et al. [32]. However, the feeding deterrent index was higher in endophytes inoculated treatments than control (Table 5).

This reduced feeding behaviour is mostly correlated with enhanced phenolics content of bacteria inoculated plants. The phenolics were reported to be toxic and thus show allelopathic effects on herbivorous insects. Studies indicate that accumulation of phenolic compounds in plants affects the growth and feeding capacity of larvae [48]. In the current study, the total phenolics content of maize leaves was higher in endophyte colonized plants over control plants (Figure 3). The result of the current study thus confirms the earlier observations of Oukala et al. [4] that tomato inoculated with endophytic *Bacillus pumilus* accumulate a greater quantity of toxic substances such as phenolics and β-1,3-glucanases against *Fusarium*. Similarly, Commare et al. [49] reported that accumulation of phenolics in tomato reduces the growth of *Helicoverba armigera.*

To the best of author’s knowledge, this is the first report of apoplastic fluid of maize, harbouring numerous plant growth promoting endophytes with bioprotective properties against the fall armyworm insect. This study resulted in isolation of 15 endophytes from both root and leaf apoplastic fluid of maize. Among those 15 endophytes, two endophytes—namely *B. amyloliquefaciens* and *Alcaligenes* sp.—were chosen as superior strains. While *B. amyloliquefaciens* inoculation significantly reduced the feeding characteristics of *S. frugiperda* on maize, *Alcaligenes* sp inoculation improved the growth of maize. Various bioassays conducted with *B. amyloliquefaciens* inoculated maize plants indicated the potential of the bacterial inoculation to reduce the leaf feeding characteristic of *S. frugiperda*, probably by enhancing feed inhibitory or toxic substances like phenolics in the plant system. It is also revealed that *B. amloliquefaciens* might be a potential bioprotective agent for *S. frugiperda.* However, the effect of the bacterial inoculation on the feeding behaviour of insects has to be studied under field conditions. Further studies are also essential to find the biochemical and physiological alterations in maize inoculated with *B. amyloliquefaciens* responsible for reduction in the feeding characteristics of *S. frugiperda*.

## Figures and Tables

**Figure 1 microorganisms-10-01850-f001:**
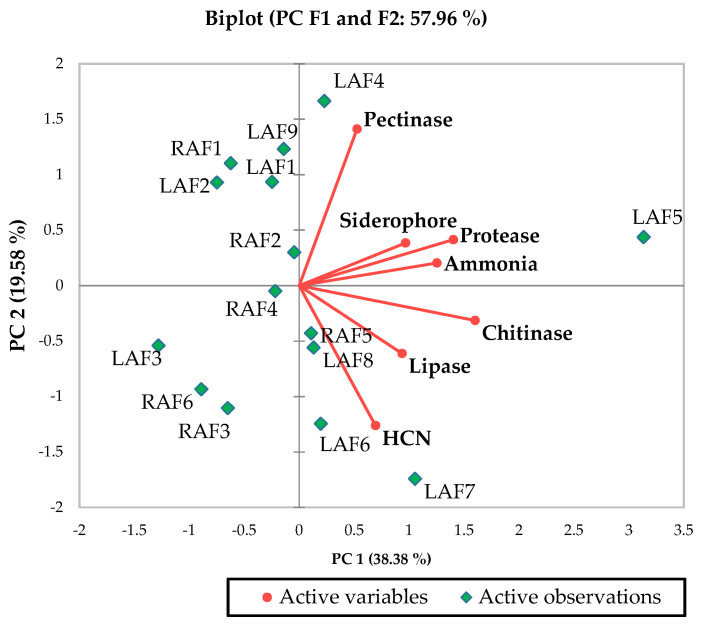
Biplot principal component analysis (PCA) describes the bioprotective potential of apoplastic fluid bacterial endophyte. LAF, leaf apoplastic fluid; RAF, root apoplastic fluid.

**Figure 2 microorganisms-10-01850-f002:**
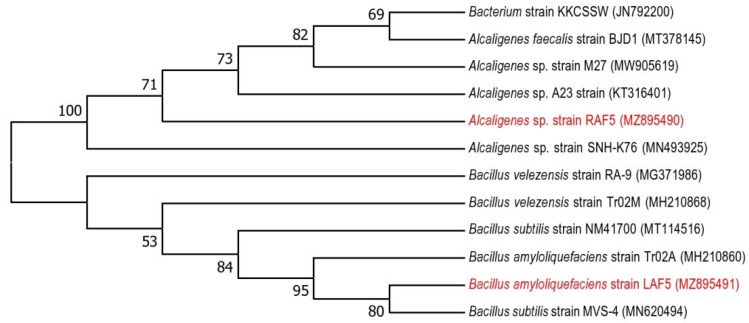
Maximum likelihood phylogenetic tree based on 16S ribosomal RNA gene sequences, showing the relationships between the bacterial taxa identified in this study. The bootstrap values ≥50% (based on 1000 replications) are shown at branching points.

**Figure 3 microorganisms-10-01850-f003:**
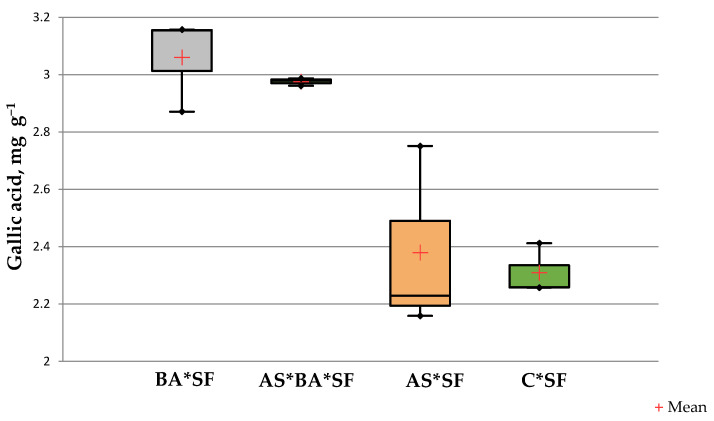
Effect of *Alcaligenes* sp. (AS)*, B. amyloliquefaciens* (BA) inoculated maize fed with *S. frugiperda* (SF) for 24 h on phenolics content. Data in figures are expressed as mean ± standard error in triplicates. C, Control.

**Figure 4 microorganisms-10-01850-f004:**
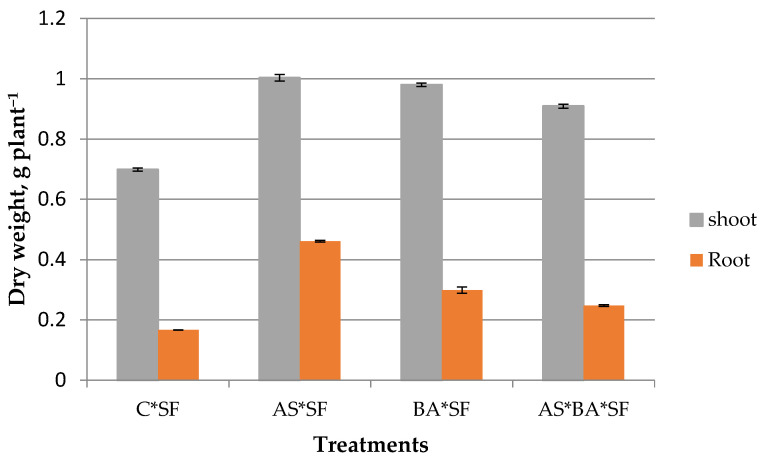
Effect of *Alcaligenes* sp. (AS)*, B. amyloliquefaciens* (BA) inoculated maize fed with *S. frugiperda* (SF) for 24 h on plant biomass production. Data in figures are expressed as mean ± standard error. C, Control.

**Table 1 microorganisms-10-01850-t001:** Plant growth promoting characteristics of maize COH6 apoplastic fluid bacterial endophytes.

Isolates	Mineral Solubilisation Index			IAA (µg mL^−1^)	GA_3_ (µg mL^−1^)
	Phosphate	Zinc	Potassium		
LAF1	2.11 ± 0.03 ^g^	nd	nd	0.25 ± 0.00 ^e^	4.25 ± 0.02 ^ed^
LAF2	3.42 ± 0.04 ^de^	2.42 ± 0.01 ^e^	4.05 ± 0.02 ^c^	0.34 ± 0.03 ^e^	26.43 ± 0.04 ^ab^
LAF3	2.24 ± 0.03 ^g^	nd	nd	0.14 ± 0.01 ^e^	1.89 ± 0.00 ^e^
LAF4	2.24 ± 0.05 ^g^	nd	nd	0.29 ± 0.01 ^e^	30.19 ± 0.04 ^a^
LAF5	3.215 ± 0.02 ^e^	nd	nd	2.15 ± 0.03 ^e^	12.67 ± 0.04d ^c^
LAF6	2.95 ± 0.02 ^f^	nd	nd	0.09 ± 0.04 ^e^	7.04 ± 0.01 ^edc^
LAF7	3.55 ± 0.02 ^d^	2.53 ± 0.02 ^d^	3.23 ± 0.01 ^d^	0.11 ± 0.02 ^e^	29.10 ± 0.02 ^a^
LAF8	4.73 ± 0.01 ^c^	nd	3.23 ± 0.02 ^d^	1.19 ± 0.01 ^e^	19.76 ± 0.03 ^abc^
LAF9	2.90 ± 0.05 ^f^	nd	nd	1.47 ± 0.02 ^e^	15.76 ± 0.01 ^abcd^
RAF1	2.95 ± 0.02 ^f^	nd	5.05 ± 0.28 ^a^	14.42 ± 0.05 ^d^	19.22 ± 0.02 ^abc^
RAF2	3.45 ± 0.02 ^de^	2.60 ± 0.02 ^c^	nd	40.59 ± 0.21 ^b^	20.01 ± 0.02 ^abc^
RAF3	5.05 ± 0.02 ^b^	2.67 ± 0.02 ^b^	2.24 ± 0.01 ^g^	1.83 ± 0.12 ^e^	21.46 ± 0.03 ^abc^
RAF4	3.35 ± 0.08 ^de^	2.31 ± 0.02 ^f^	2.44 ± 0.01 ^f^	27.15 ± 0.03 ^c^	21.89 ± 0.01 ^abc^
RAF5	10.75 ± 0.14 ^a^	3.51 ± 0.05 ^a^	4.95 ± 0.03 ^b^	58.04 ± 0.07 ^a^	30.61 ± 0.02 ^a^
RAF6	3.60 ± 0.23 ^d^	2.66 ± 0.03 ^b^	2.75 ± 0.03 ^e^	1.87 ± 0.05 ^e^	26.25 ± 0.01 ^ab^

Values with different letters are significantly different according to Duncan’s multiple range test (DMRT) (*p* ≤ 0.05). LAF, leaf apoplastic fluid; RAF, root apoplastic fluid; nd, no detected activities.

**Table 2 microorganisms-10-01850-t002:** Bioprotective properties of maize apoplastic fluid bacterial endophytes.

Isolates	Siderophore (%)	HCN (OD Value)	Ammonia (µg mL^−1^)	Hydrolytic Enzymes			
				Lipase (cm)	Protease (U mL^−1^)	Pectinase (U mL^−1^)	Chitinase (U mL^−1^)
LAF1	nd	nd	0.59 ± 0.02 ^abc^	0.70 ± 0.01 ^b^	nd	6.89 ±0.04 ^a^	0.17± 0.02 ^d^
LAF2	4.74 ± 0.001 ^g^	0.04 ± 0.02 ^f^	nd	nd	nd	6.38 ± 0.09 ^c^	nd
LAF3	nd	0.01 ± 0.08 ^j^	0.12 ± 0.02 ^c^	nd	nd	3.24 ± 0.06 ^k^	nd
LAF4	10.36 ± 0.01 ^b^	0.03 ± 0.08 ^i^	0.12 ± 0.07 ^abc^	nd	301.05 ± 0.03 ^b^	6.33 ± 0.03 ^c^	nd
LAF5	1.09 ± 0.04 ^e^	0.12 ± 0.02 ^c^	1.68 ± 0.03 ^a^	0.70 ± 0.06 ^b^	386.05 ± 0.01 ^a^	6.65 ± 0.02 ^b^	3.44 ± 0.07 ^a^
LAF6	4.32 ± 0.01 ^c^	0.16 ± 0.02 ^e^	0.66 ± 0.06 ^abc^	0.61± 0.06 ^c^	51.05 ± 0.01 ^d^	4.67 ± 0.014 ^f^	nd
LAF7	2.88 ± 0.02 ^f^	0.13 ± 0.05 ^b^	0.40 ± 0.01 ^bc^	0.81 ± 0.06 ^a^	241.05 ± 0.05 ^c^	3.68 ± 0.02 ^i^	2.05 ± 0.02 ^b^
LAF8	2.33 ± 0.04 ^d^	0.13 ± 0.05 ^bc^	0.93 ± 0.03 ^abc^	nd	nd	5.09 ± 0.02 ^e^	1.16 ± 0.02 ^c^
LAF9	1.55 ± 0.03 ^e^	0.04 ± 0.02 ^g^	0.53 ± 0.01 ^abc^	nd	211.05 ± 0.02 ^c^	6.44 ± 0.02 ^c^	nd
RAF1	1.22 ± 0.01 ^e^	nd	0.81 ± 0.03 ^abc^	nd	nd	5.95 ± 0.08 ^d^	nd
RAF2	12.29 ± 0.04 ^a^	0.03 ± 0.02 ^h^	1.09 ± 0.02 ^abc^	nd	nd	3.91 ± 0.02 ^h^	nd
RAF3	2.68 ± 0.03 ^d^	0.11 ± 0.08 ^d^	nd	0.40 ± 0.08 ^d^	nd	4.24 ± 0.01 ^g^	nd
RAF4	2.93 ± 0.00 ^d^	nd	0.90 ± 0.08 ^abc^	0.70 ± 0.02 ^b^	nd	4.53 ± 0.09 ^fg^	nd
RAF5	10.07 ± 0.08 ^b^	0.14 ± 0.02 ^a^	0.95 ± 0.04 ^abc^	nd	nd	4.62 ±0.07 ^fg^	nd
RAF6	1.14 ± 0.14 ^e^	0.13 ± 0.05 ^b^	nd	nd	nd	4.45 ± 0.02 ^g^	nd

Values with different letters are significantly different according to Duncan’s multiple range test (DMRT) (*p* ≤ 0.05). LAF, leaf apoplastic fluid; RAF, root apoplastic fluid; nd, no detected activities.

**Table 3 microorganisms-10-01850-t003:** Effect of *B. amyloliquefaciens* inoculated maize leaves as feed on the growth of *S. frugiperda* in no-choice bioassay.

Treatment	RGR (mg mg^−1^ day^−1^)	RCR (mg mg^−1^ day^−1^)	ECI (%)	ECD (%)	MC (%)	FDI (%)
BA−	1.29 ± 0.16	15.63 ± 2.45	8.26 ± 1.73	21.88 ±1.42	78.12 ± 0.56	0.00
BA+	1.00 ± 0.19	7.16 ± 3.48	13.97 ± 6.35	49.29 ± 2.01	50.71 ± 0.29	42.64 ± 2.64
MD	0.29	8.47	5.74	27.41	27.41	42.64
*p* (f test)	0.24	3.15	6.62	4.72	3.47	7.71
T	18.65	216.27	36.51	145.66	25.33	587.88
*p*	0.001	0.001	0.042	0.001	0.000	0.032

Statistical significance (*p* ≤ 0.05) was calculated by independent sample *t*-test. BA−, without *B. amyloliquefacines*; BA+, with *B. amyloliquefacines*; MD, mean difference; RGR, relative growth rate; RCR, relative consumptive rate; ECI, efficiency of conversion of ingested food; ECD, efficiency of conversion of digested food; MC, metabolic cost; FDI, feeding deterrent index.

**Table 4 microorganisms-10-01850-t004:** Effect of *B. amyloliquefacines* on the growth of *S. frugiperda in* larval dip bioassay.

Treatment	RGR (mg mg^−1^ day^−1^)	RCR (mg mg^−1^ day^−1^)	ECI (%)	ECD (%)	MC (%)	FDI (%)
BA−	2.10 ± 0.29	4.44 ± 0.97	47.28 ± 1.37	29.76 ± 8.58	70.24 ± 8.58	0.00
BA+	1.69 ± 0.38	2.72 ± 0.39	62.41 ± 6.44	46.56 ± 7.54	53.44 ± 7.54	17.70 ± 6.62
MD	0.41	1.72	15.13	16.80	16.80	17.70
*p* (f test)	2.59	1.89	0.05	3.37	7.41	7.71
T	18.52	22.04	48.26	37.54	18.50	86.45
*p*	0.02	0.001	0.042	0.001	0.001	0.032

Statistical significance (*p* ≤ 0.05) was calculated by independent sample *t*-test. BA−, without *B. amyloliquefacines*; BA+, with *B. amyloliquefacines*; MD, mean difference; RGR, relative growth rate; RCR, relative consumptive rate; ECI, efficiency of conversion of ingested food; ECD, efficiency of conversion of digested food; MC, metabolic cost; FDI, feeding deterrent index.

**Table 5 microorganisms-10-01850-t005:** Impact of *B. amyloliquefaciens* and *Alcaligenes* sp. inoculated maize leaves as feed on the growth of *S. frugiperda* (detached leaf bioassay).

Treatment	RGR (mg mg^−1^ day^−1^)	RCR (mg mg^−1^ day^−1^)	ECI (%)	ECD (%)	MC (%)	FDI (%)
C*SF	1.27 ± 0.46 ^b^	9.55 ± 5.63 ^a^	13.33 ± 6.75 ^d^	8.16 ± 6.13 ^a^	91.84 ± 1.37 ^c^	0.00 ^d^
AF*SF	1.63 ± 0.31 ^a^	6.63 ± 1.36 ^b^	24.53 ± 1.26 ^b^	5.95 ± 0.22 ^c^	94.05 ± 0.22 ^ab^	45.21 ± 1.10 ^b^
BA*SF	0.60 ± 0.15 ^d^	2.09 ± 0.49 ^d^	30.00 ± 6.13 ^a^	3.89 ± 0.17 ^d^	96.11 ± 0.17 ^a^	68.00 ± 2.69 ^a^
AS*BA*SF	1.00 ± 0.20 ^c^	5.39 ± 1.08 ^c^	18.52 ± 0.14 ^c^	6.54 ± 0.19 ^b^	93.46 ± 0.19 ^ab^	32.08 ± 1.60 ^c^
F	1.13	1.29	561.15	649.24	2.99	7.86
*p*	0.04	0.001	0.000	0.01	0.073	0.000

Values with different letters are significantly different according to Duncan’s multiple range test (DMRT) (*p* ≤ 0.05). RGR, relative growth rate; RCR, relative consumptive rate; ECI, efficiency of conversion of ingested food; ECD, efficiency of conversion of digested food; MC, metabolic cost; FDI, feeding deterrent index; C, Control; SF, *S. frugiperda*; AS, *Alcaligenes* sp.; BA, *B. amyloliquefaciens.*

## Data Availability

Raw sequence data reported in this paper have been deposited in the NCBI GenBank under accession number MZ895490 and MZ895491.

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
