# Peer review of "Maize Apoplastic Fluid Bacteria Alter Feeding Characteristics of Herbivore (Spodoptera frugiperda) in Maize"

_microorganisms, 2022, doi:10.3390/microorganisms10091850_

Round 1
Reviewer 1 Report
Authors presented the role of endophytic bacteria of maize root and leaf apoplastic fluid against S. frugiperda and plant growth promoting properties. The study proved the the potential utility of Alcaligenes sp. and B. amyloliquefaciens for improving growth and biotic (S. frugiperda) stress tolerance in maize, which was determined by isolating and screening potential plant growth promoting endophytes from apopastic fluid. The bioprotective potential of endophytes was evaluated through bioassays and by assessing selected physiological characteristics of maize. The methods presented are clear, repeatable and possible to implement. The manuscript is well written and structured, therefore I recommend its publication Nevertheless, the manuscript needs to be corrected. Changes and suggestions are described in the attached pdf file.

Author Response
Sir/Madam
Greetings. The article has been changed as per your suggestions. The author’s response for reviewer’s queries is given in attached file

Reviewer 2 Report
The manuscript reports the isolation of various bacteria within the corn apoplast and the search among these for strains with PGPR activity and, in particular, that counteracted the attacks of Spodoptera frugiperda. After having isolated 15 strains, the manuscript reports numerous experiments to identify the strains with greater biocontrol activity towards Spodoptera larvae and plant growth stimulation.
The authors report the selection of a Bacillus amyloliquefaciens strain with a moderate larval contrast activity in plant attack experiments, and of an Alcaligenes strain with good plant stimulation ability.
The work is interesting and original: there are not many works in the literature on bacterial biocontrol against the very fearsome and destructive attacks of Spodoptera larvae.
In Figure 1: I suggest combining both the principal component variables and observations in a single biplot display, the reading would be much clearer. PCA is the acronym of Principal Component Analysis. Please change.
I do not understand in figure 4 the letters that are above the columns of the histograms. If different letters indicate different statistical values there is something wrong, it is also noticeable visually. Please explain and add an explanation in the caption.
Minor suggestions
Line 27: please use international measures, ha.
Lines 34-35: you say that plants are associated with beneficial and deleterious microorganisms. A plant is associated with beneficial microorganisms, you cannot call deleterious microorganisms associated.
Line 65: add “India” for the affiliation
Line 75: change with “the samples were placed inside…”
Lines 89 and 92: change to “Salkowski reagent”
Paragraphs from 2.3.4 to 2.3.7: review the numbers of the bibliography, they seem wrong.
Line 182: leaves cannot be surface sterilized with sterile distilled water!
Line 207: You introduce Alkaligenes without explaining that it is one of the isolates from the maize apoplast. Please add a phrase to explain.
Lines 219-221. Please detail how the experiment was conducted, if the leaves were inoculated, how and how much.
Line 239: change to: “Principal Component Analysis”
Lines 281-282: the name of the bacteria in italics.
Author Response
Sir/Madam
Greetings. The article has been changed as per your suggestions. The author’s response for reviewer’s queries is given in attachment file.
